# Scaling laws in jet classification

Joshua Batson° and Yonatan Kahn[1,2,3]⋆

**1** Center for Artificial Intelligence Innovation and Department of Physics,
University of Illinois Urbana-Champaign, Urbana, IL USA
**2** Department of Physics, University of Toronto, Toronto, Canada
**3** Vector Institute, Toronto, Canada

⋆ yf.kahn@utoronto.ca

## Abstract

We demonstrate the emergence of scaling laws in the benchmark top versus QCD jet classification problem in collider physics. Six distinct physically-motivated classifiers exhibit power-law scaling of the binary cross-entropy test loss as a function of training set size, with distinct power law indices. This result highlights the importance of comparing classifiers as a function of dataset size rather than for a fixed training set, as the optimal classifier may change considerably as the dataset is scaled up. We speculate on the interpretation of our results in terms of previous models of scaling laws observed in natural language and image datasets.


## Contents

° Now employed at Anthropic.

# 1  Introduction

In just the past few years, neural scaling laws [1] have gained prominence both as an emergent property of large machine learning (ML) models and as a practical tool for predicting the performance of such models. Across a wide variety of tasks and architectures, the training or test loss $L$ is observed to follow a power law,

$$L(T) = AT^{-\alpha_T} + C \,, \tag{1}$$

where $T$ represents a variable such as the size of the training set, the number of parameters, or the amount of compute; $\alpha_T$ is a task-dependent power law index which depends on the choice of variable $T$ but only weakly on architecture and other hyperparameters; and $C$ is the irreducible loss which persists in the limit of infinite data/parameters/compute. A first-principles understanding of the robustness and ubiquity of these power laws remains elusive (though see [2–4]), but in industry applications, the scaling of performance is so reliable that it can be used to correctly predict the trained model loss after scaling the model up by multiple orders of magnitude [5,6].

ML models have also gained prominence as tools for solving problems in physics. A common application is the processing of data from high-energy particle colliders (see the "living review" [7] for a continually-updated compendium of references). In such experiments, the volume of data is enormous, even by industry standards: the Large Hadron Collider (LHC) generates about 1 petabyte per *second* of raw data [8], the vast majority of which must be discarded in order to store the (still enormous) 160 petabytes per year of "interesting" data to disk. The upgraded High Luminosity LHC (HL-LHC) expects to increase both the total event rate and the recorded data rate by more than an order of magnitude [9,10]. If some fraction of the discarded data could be used to train ML models to aid with analysis tasks, that would have enormous practical implications for the design of the hardware "trigger" that determines which events to keep for offline analysis. While there is some concern among builders of large language models that such models might "run out of data" before saturating the returns to scale, ML models in physics are currently trained on a vanishingly small fraction of the total data available.[1] In spite of this abundant resource, in most collider physics studies using ML tools, competing models are often compared to one another at some fixed but arbitrary $T$.

In this paper, we demonstrate empirically that scaling laws also emerge in physics tasks. Using the benchmark example of discriminating two classes of jets (sprays of collimated particles detected at high-energy colliders) in simulated data [11], we show that six different physically-motivated classifiers yield widely-varying power laws, extending over 3–4 orders of magnitude in training set size $T$ (see Fig. 1). Due to the large variation in $\alpha_T$ among classifiers, the "best" classifier may change as a function of $T$. This phenomenon has only recently begun to be pointed out (see Refs. [12,13] for example) and does not yet seem to be widely appreciated. Consequently, we urge physicists developing ML models to present their results in log-log space as a function of training set size, both to explore the extent to which power laws are ubiquitous in physics applications (and perhaps to use physics tools to derive the power law index), but also to understand the benefit of scaling up the training set compared to other models. Throughout this paper, we endeavor to use non-technical language which we hope will be comprehensible to both physicists and non-physicists.

---

[1]This observation holds whether or not the training dataset is drawn from simulated data or real data. Independent of the subtle question of whether simulations are accurately capturing the true distribution of real data, the only limitation to scaling up the training sets generated from Monte Carlo event generators seems to be storage space, rather than any principled reason.

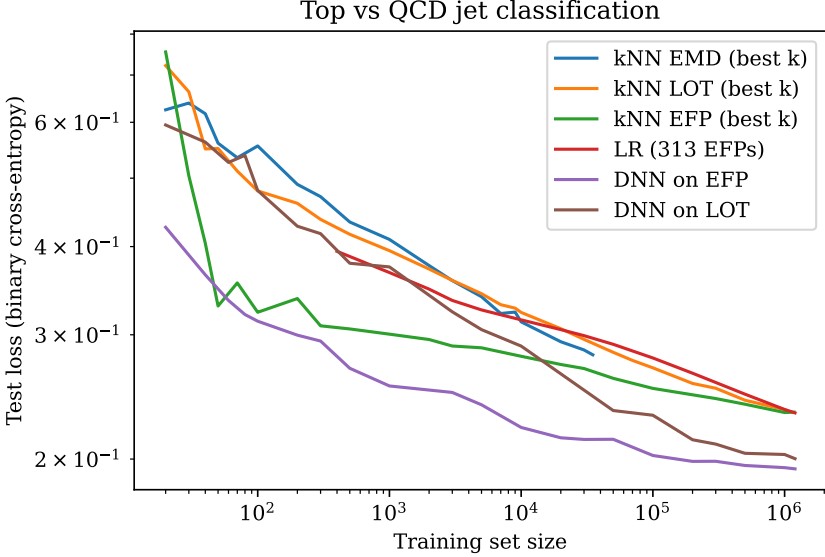

Figure 1: Performance of various classifiers trained to distinguish top quark jets from light-quark and gluon QCD jets, as a function of training set size. The classifier architectures and hyperparameters are discussed in Sec. 4. Power law behavior is evident over several decades for all six classifiers we consider. The rank ordering among these six classifiers changes five times as the training set size is increased. If the observed power laws extend, the ordering may change again at larger training set size.

# 2 Classification task: Top versus QCD jet discrimination

Here we briefly review the setup of the benchmark classification problem of Ref. [11]. Quarks are fundamental particles which feel the strong nuclear force, but they are never observed as free particles because of a property of quantum chromodynamics (QCD) called confinement. Rather, they manifest at particle detectors as a large number of particles clustered around a particular direction – a jet. Identifying the primary quark which initiated the jet is a key analysis task at high-energy colliders, since different physical processes of interest may create different quarks. The heaviest quark is the top quark, which decays immediately to three quarks such that each of its decay products forms its own "subjet" within the top quark jet. The lightest quarks are copiously produced at colliders in events where nothing much of interest happens, and thus these "QCD jets" can pose a background for searches for rarer processes.

The classification task is therefore to distinguish a top jet from a QCD jet, once some "jet algorithm" (see [14] for a pedagogical overview) has clustered the detected particles into jets. The data simply consists of the energies $E_i$ and momenta $\vec{p}_i$ of the particles in the jets, which can be viewed as an unordered point cloud and is conveniently expressed as a list of energy-momentum 4-vectors $p_i = (E_i, \vec{p}_i)$. In the benchmark dataset of Ref. [11], both classes of simulated jets have up to 200 constituents, with zero-padding used to balance the size of the arrays for jets with fewer particles. Early attempts to harness deep learning tools for jet classification mapped these 4-vectors into a "jet image" [15–17] where the pixels of the image represent a discretization of the angular coordinates and the grayscale intensity of the pixel is the total energy in that angular region. Using jet image space to visualize the classes, a top jet is "three blobs" and a QCD jet is "mostly one blob." However, this mapping into jet image space obscures the point cloud nature of jets, so we will focus on classifiers which take functions of the 4-vector point clouds as input.

We choose to focus on the particular classification problem of top versus QCD jets because it is the benchmark jet classification problem which is expected to have the smallest irreducible loss. In autoregressive language models, the irreducible loss can be attributed to the entropy of language [18]. For classification problems in physics, the irreducible loss may arise from quantum mechanics, which implies that there may be jets which cannot be assigned to a distinct class even in principle. This issue is most apparent in a related problem of distinguishing quark jets from jets initiated by gluons, the particles which mediate the strong force. While it has been shown that there are some quark jets which are fully distinguishable from gluon jets [19,20], such jets make up a set of measure zero of the data manifold, leading to a large irreducible loss when sampling random jets. This makes it difficult to identify a robust power law over several decades of $T$ before the saturation of the irreducible loss takes over.

## 3 Physically-motivated classifiers

### 3.1 Infrared and collinear safety

QCD is a relativistic quantum field theory, which means that its predictions are probabilistic and covariant under the linear transformations of space and time coordinates allowed by special relativity. In the regime where observables are calculable as a perturbative series, the combination of relativity and quantum mechanics implies that certain quantities are ill-defined because probabilities become singular in the infrared and collinear limits. Infrared (or soft) singularities arise from a particle emitting another particle whose energy is vanishingly small. Collinear singularities arise from a particle splitting into two particles, each carrying a fraction of the energy of the original particles but with their momenta exactly aligned. Observables that are insensitive to soft-particle emission or collinear splitting are called infrared and collinear (IRC) safe. The canonical example of a *non*-IRC safe observable is total particle number, since this is not robust to soft or collinear emission: two jets which differ only by the addition of an arbitrarily low-energy particle, or by splitting one particle into two collinear ones, will have particle numbers that differ by one but are otherwise indistinguishable in perturbation theory. By contrast, the total energy of the jet is an IRC-safe observable.

The relationship between IRC-safe observables and calculable observables is quite subtle [21,22], and it is known that IRC-unsafe information can improve classification performance (see e.g. [23]). Nonetheless, IRC-safe observables have a satisfying interpretation in terms of the geometry of the data manifold [22]. Since several studies of scaling laws have noted the possible relationship of the power law slope to the intrinsic or extrinsic dimension of the data [2–4], we focus exclusively on IRC-safe classifiers in this work. Our goal is explicitly *not* to identify the overall best-performing classifier, but instead to emphasize that such an assessment always depends on the size of the training set.

### 3.2 Energy flow polynomials (EFP)

Energy flow polynomials (EFPs) [24] are an overcomplete linear basis for all IRC-safe jet observables. Let $z_i$ be the fraction of the total jet energy carried by particle $i$, and $\theta_{ij}$ be the angular distance between particles $i$ and $j$ in a chosen angular coordinate system (for example, in spherical coordinates, $\theta_{ij} = \sqrt{(\theta_i - \theta_j)^2 + (\phi_i - \phi_j)^2}$). The EFPs are in one-to-one correspondences with loopless multigraphs $G$, and for $G$ with $N$ vertices and edges $(k, \ell)$, the associated EFP is defined as

$$\text{EFP}_G = \sum_{i_1=1}^{M} \cdots \sum_{i_N=1}^{M} z_{i_1} \cdots z_{i_N} \prod_{(k,\ell) \in G} \theta_{i_k i_l}, \tag{2}$$

where $M$ is the number of particles in the jet. This definition is manifestly permutation-symmetric and therefore respects the point cloud nature of the jet, and Ref. [24] proves that it is IRC-safe. Connected multigraphs with more edges have a higher degree of the angular monomial, and roughly speaking, probe smaller angular scales. A reasonable truncation scheme for the EFPs therefore restricts to multigraphs with at most $d$ edges (and consequently at most degree $d$ in the angular factor): there are 313 nontrivial EFPs for $d \leq 6$ and 999 EFPs for $d \leq 7$.[2] Computing the EFPs of $d \leq 6$ on a jet consisting of $M$ 4-vectors is thus a nonlinear map from $\mathbb{R}^{4M}$ to $\mathbb{R}^{313}$, for example, which can be viewed as a physically-motivated preprocessing of the data into a useful feature space.

### 3.3 Energy mover's distance (EMD)

Rather than preprocessing the jets by mapping to Euclidean space, we can define an IRC-safe distance measure directly on the space of jets. This is known as the energy mover's distance (EMD) [22, 25], which is a modification of the earth mover's distance used in computer vision [26]. Unlike the dimensionless EFPs, it has a natural dimensionful scale (with units of energy) and is defined using the energies $E_i$ of the particles in the jet rather than just the energy fractions. For events $\mathcal{E}$ and $\mathcal{E}'$, the EMD is defined as the solution to an optimal transport problem for coefficients $f_{ij}$,

$$\text{EMD}(\mathcal{E}, \mathcal{E}') = \min_{\{f_{ij}\}} \sum_{ij} f_{ij} \left( \frac{\theta_{ij}}{R} \right)^{\beta} + \left| \sum_i E_i - \sum_j E'_j \right|, \tag{3}$$

subject to the constraints

$$f_{ij} \geq 0, \qquad \sum_j f_{ij} \leq E_i, \qquad \sum_i f_{ij} \leq E'_j, \qquad \sum_{ij} f_{ij} = \min(E, E'), \tag{4}$$

where $E$ and $E'$ are the total energies of $\mathcal{E}$ and $\mathcal{E}'$, respectively. The angular distance $\theta_{ij}$ is identical to that used in the EFP definition; the angular scale $R$ and the angular exponent $\beta$ are hyperparameters of the EMD. The EMD is a proper metric satisfying the triangle inequality for $\beta = 1$, and we will restrict to this value from now on. Ref. [22] proves that two events are separated by zero EMD iff they differ by the addition of zero-energy particles or exactly collinear splittings, thus providing a direct connection to IRC safety.

### 3.4 Linearized optimal transport (LOT)

The EMD has a very clean physical and mathematical interpretation, but as a nonlinear optimal transport problem, it is quite expensive to compute: even with modern optimal transport libraries, computing the $\binom{T}{2}$ entries of the EMD matrix for $T > 10^5$ takes weeks of multi-core CPU or GPU walltime.[3] The EMD with $\beta = 1$ is an example of a $p$-Wasserstein distance with $p = 1$, which (like the $\ell^1$ norm) does not have a pseudo-Riemannian structure, but modifying the EMD to a 2-Wasserstein distance, which is defined in terms of an $\ell^2$ norm, admits a linearization which embeds the events $\mathcal{E}$ into Euclidean space [28]. Using this linearized optimal transport (LOT) approximation, which projects onto the tangent plane of the 2-Wasserstein pseudo-Riemannian manifold at a given point, the distance between events is approximated by the $\ell^2$ Euclidean distance on the tangent plane. The total computational cost of LOT is then $\mathcal{O}(T)$ for the expensive 2-Wasserstein distances, and $\mathcal{O}(T^2)$ only for the vastly more efficient Euclidean distances.

---

[2]Note that the first EFP with $d = 0$ is trivial, since it is just $\sum_i z_i = 1$, so we will exclude this from our analysis.
[3]Though, see Ref. [27] for a much faster approximation.

The specific LOT algorithm defined in Ref. [28] is as follows. First, define a reference jet $\mathcal{R}$ containing $M_R$ particles of energies $R_i$ and angular positions $X_i$ (i.e. $(\theta_i, \phi_i)$ in spherical coordinates). For another event $\mathcal{E}$ with energies $E_j$ and angular positions $x_j$, let $f_{ij}$ denote the optimal transport plan for the 2-Wasserstein distance

$$W_2(\mathcal{R}, \mathcal{E}) = \min_{\{f_{ij}\}} \sqrt{\sum_{i=1}^{M_R} \sum_{j=1}^{M} f_{ij} ||X_i - x_j||^2}, \tag{5}$$

subject to the constraints in Eq. (4). For simplicity we have assumed that the energy of all events $\mathcal{E}$ (as well as $\mathcal{R}$) is normalized to 1, so the energy cost factor in Eq. (3) vanishes. The average locations of the energy transport for the particles in $\mathcal{R}$,

$$z_i := \frac{1}{R_i} \sum_{j=1}^{M} f_{ij} x_j, \tag{6}$$

define a map from $\mathcal{E}$ to $\mathbb{R}^{M_R}$. While for any finite $M_R$, LOT does not define a metric on the space of events, in the continuum limit $M_R \to \infty$, Ref. [28] proves that the LOT does converge to a true metric, which is not isomorphic to the EMD metric. The ideas behind LOT can be generalized to events with different energy, where the energy cost factor $\left| \sum_i E_i - \sum_j E_j' \right|$ is important for classification, using a linearization of the Hellinger-Kantorovich distance [29]. However, since the top/QCD classification task is mostly a question of differing angular distributions rather than total energy, we will focus on the behavior of classifiers based on LOT as defined above.

# 4 Architecture and hyperparameter details

We now define several classifiers, both parametric and non-parametric, whose inputs are top and QCD jet events processed using the EFPs, EMD, and/or LOT. We briefly motivate each choice of classifier, and describe which aspects of the geometry of the data we expect it to measure, as well as how it relates to the other classifiers. For all classifiers, we define the loss as the binary cross-entropy,

$$L = -\frac{1}{T_{\text{test}}} \sum_{i=1}^{T_{\text{test}}} \left( y_i \log p_i + (1 - y_i) \log(1 - p_i) \right), \tag{7}$$

where $y_i = 0\,(1)$ for QCD jets (top jets), $p_i$ is the classifier's probability that event $i$ in the test set is a top jet, and $T_{\text{test}}$ is the size of the test set. For all classifiers except the one using the EMD, we use $T_{\text{test}} = 4 \times 10^4$; due to computational limitations with the EMD, we use $T_{\text{test}} = 10^4$. We train each classifier on training sets of various size $T$, obtained by selecting a random sample from the full training set of size $1.2 \times 10^6$.

1. **kNN EMD**. Since the EMD directly measures distances between events in an IRC-safe manner, we define a $k$-nearest neighbors (kNN) classifier with respect to the EMD distance matrix using the `KNeighborsClassifier` method in `scikit-learn` [30]. Before computing the EMDs, we preprocess the jets by centering them in the angular coordinates such that the transverse-momentum-weighted centroid lies at the origin; this reduces the effective dimensionality of the dataset by removing translations in angular space, such that the dimensions which remain are those which differentiate the jets from

one another based on the relative positions and energies of their constituents. As mentioned earlier, computing the EMD matrix for large $T$ is intractable, so we only consider $T$ up to $3.5 \times 10^4$ which can be computed within a 24-hour walltime on a cluster. We treat the number of nearest neighbors $k$ as a hyperparameter, which we optimize over using a validation set of size 5000. Ref. [25] used $k = 32$ for fixed $T$, but we find that for $T > 100$, the optimal value of $k$ is $\mathcal{O}(50)$ (see Fig. 7 in Appendix A below). We take $R = 0.8$ which is the jet radius for the jet algorithm used in the training and test sets, and discuss the result of changing the value of $R$ in Appendix A.

2. **kNN LOT**. Alternatively, we can perform kNN classification using the Euclidean metric on $\mathbb{R}^{M_R}$ provided by the LOT embedding. One might expect that this classifier will perform similarly, if not identically, to EMD for sufficiently large $M_R$, in the sense that if the classifier is based purely on distances, the Euclidean distance on the tangent space is very close to the true metric distance on the manifold so long as events are close together. Furthermore, the computational efficiency of LOT permits us to compute the LOT distance matrix on the full training set with $T = 1.2 \times 10^6$. Indeed, this classifier was compared to EMD-based classifiers in Ref. [28] for fixed $T$. We optimize over $k$ using a validation set of size $10^4$, and find, consistent with Ref. [28], that optimal $k$ values for sufficiently large $T$ are $\mathcal{O}(50)$. The size $M_R$ of the reference jet is also a hyperparameter for this classifier, but we will not optimize over it; rather, we show in Appendix A that different choices (taking the particles in the reference jet to be equally spaced in rapidity-azimuth space with equal energies, as in [28]) yield identical losses.

3. **kNN EFP**. The LOT embedding does not necessarily preserve the IRC-safe properties of the EMD. However, we can also treat the EFP coefficients as (an infinite set of) coordinates on the jet manifold which do respect IRC safety. Choosing a finite set of EFPs amounts to mapping the jet manifold into a finite-dimensional Euclidean space, where we can perform kNN classification using the standard Euclidean distance matrix. The number of EFPs is a hyperparameter for this classifier: as a starting point for our study, we will take the first 313 nontrivial EFPs in the ordering of Ref. [24], corresponding to angular polynomials with degree 6 or less. We optimize over $k$ using the same validation set as for kNN LOT.

4. **Logistic regression (LR) on EFPs**. Since the EFPs form a linear basis for IRC-safe observables, a linear classifier, such as logistic regression, should suffice to determine a fairly good decision boundary. Such an LR classifier was also studied in Ref. [24], though not as a function of dataset size. Despite the fact that the EFP basis is infinite, Ref. [24] showed that linear classifiers based on the first 1000 EFPs achieved comparable performance to jet image-based convolutional neural networks (CNNs) despite the vastly smaller number of model parameters. We use the `scikit-learn` logistic regression method with the `liblinear` solver on the first 313 nontrivial EFPs, but since linear methods without regularization exhibit catastrophic overfitting for $T$ smaller than the number of features, we restrict to $T \geq 400$. We illustrate the result of varying the number of EFPs in the regression in Fig. 3.

5. **DNN on EFPs**. The EFPs span the space of functions on the IRC-safe event manifold with metric given by the EMD, so one might expect a *nonlinear* combination of a finite number of EFPs to have sufficient expressivity to define an optimal decision boundary directly on the event manifold. To that end, we train a deep neural network (DNN) classifier on the same 313 EFP features used in the kNN and LR classifiers described above. We do not attempt a full hyperparameter scan, but rather choose hyperparameters and architectures similar to Ref. [24] in order to make contact with previous results

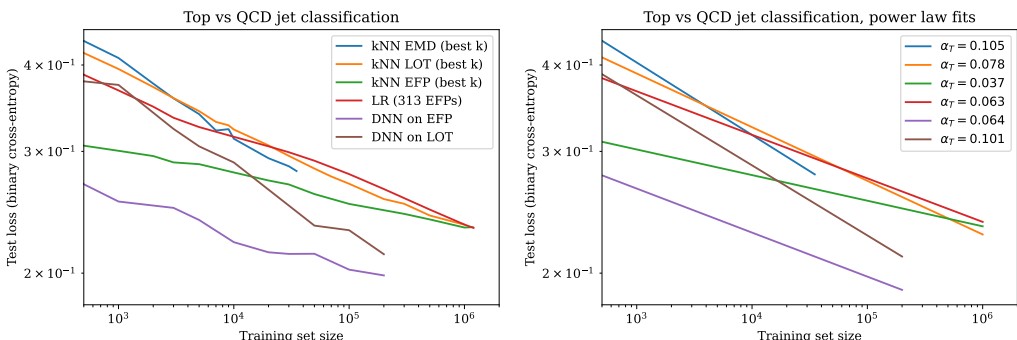

Figure 2: Comparison of classifier loss (left) and power law fits (right) for various classifiers, assuming zero irreducible loss ($C = 0$). Most of the classifiers exhibit noise and/or broken power laws with a different slope at small $T$, so in the left panel we show the same data as in Fig. 1 restricted to $T \geq 500$ to study the large-$T$ behavior. Likewise, a saturation of the loss for the DNN classifiers may begin at sufficiently large $T$, so we only show the losses before the approach to the loss floor; see Fig. 5 below for the fit including the loss floor. In the right panel we show the power law fits over the same range of $T$.

in the literature: 3 fully-connected hidden layers of width 128, ReLU activations with critical weight and bias initializations [31], softmax output, Adam [32] optimizer with a learning rate of $5 \times 10^{-4}$, batch sizes of 16 for $T \leq 300$ and 128 for $T > 300$, and 200 epochs of training with early stopping on a validation set of size $10^4$ using a patience of 20. We verified that all networks achieved early stopping within the 200 epochs of training. Based on the robustness of scaling laws to hyperparameter choices observed in language models, we do not expect our results to change significantly with different DNN hyperparameters; as a preliminary check, we verified that tripling both the width and depth of the network does not improve the loss for the largest values of $T$, even though such scaled-up models overparameterize the data.

6. **DNN on LOT**. Finally, we consider training a DNN on the Euclidean embedding of the events given by LOT. This classifier does not have a particularly transparent geometric interpretation, but intuitively, one might expect that a DNN could learn to "undo" the linearization of the 2-Wasserstein metric and thereby improve performance compared to the kNN LOT classifier as the size of the training set increases. We use the same hyperparameters and architecture as for the EFP DNN.

For small $T$, we average the results of each classifier multiple times over different random training sets in order to reduce noise: for logistic regression we average 10 times for $T \leq 10^5$, for the kNN classifiers we average 10 times for $T \leq 5000$, and for the DNN classifiers we average 5 times for $T \leq 300$.

# 5 Results

Fig. 1 summarizes the results of the trained classifiers described in Sec. 4 above. While the behavior of all classifiers at small $T$ is still noisy despite averaging, clear power laws are visible over several decades of $T$. We note that the rank ordering of the classifiers changes as the training set size increases. While DNN on EFP remains the leader throughout, at $T = 10^3$, kNN on EFPs is a close second, LR on EFPs ties DNN on LOT, and kNN using EMD is in last

Table 1: Power-law slopes and $1\sigma$ error bars from the fits in Fig. 2.

| $\alpha_T$ | Classifier |
|---|---|
| $0.105 \pm 0.003$ | kNN EMD |
| $0.101 \pm 0.004$ | DNN on LOT |
| $0.078 \pm 0.002$ | kNN LOT |
| $0.063 \pm 0.001$ | LR on EFP |
| $0.064 \pm 0.004$ | DNN on EFP |
| $0.037 \pm 0.001$ | kNN EFP |

place. By $T = 3 \times 10^4$, DNN on LOT pulls into second place, kNN using EMD has beat both kNN on LOT and LR on EFP and is on track to beat kNN on EFP, and LR on EFPs is in last place. Depending on how one extrapolates the two DNN models (as we discuss further below), it seems possible that at $T = 10^7$, either DNN on LOT or kNN on EMD would have the lowest loss of all.

The performance of a classifier depends on its performance at small $T$, the slope of the scaling law, and its leveling off due to the irreducible loss (if present). To study the slope in more detail, we restrict to $T \geq 500$ and perform a linear least-squares fit of the losses to the logarithm of Eq. (1) with $C = 0$.[4] Since the slopes of the DNN classifiers appear to soften at large $T$, we also exclude $T > 2 \times 10^5$ from the fit for those classifiers, though we investigate below whether this behavior may result from approaching a nonzero irreducible loss floor.

The results of the power law fits are shown in Fig. 2, with the classifiers sorted by power law slope in Table 1, including $1\sigma$ error bars on the fit. Of course, there are other sources of error (including the random sampling of training sets, and initialization variance in the DNNs), so these errors are only a lower bound. Nonetheless, the range of slopes which are *not* within each others' error bars is remarkable, implying (as noted above) that the relative ordering of the classifier losses changes multiple times as the size of the training set is scaled up. The shallowest slopes are obtained by the three EFP classifiers, meaning they exhibit smaller marginal returns to additional data in spite of starting with the lowest losses at $T = 500$. The slopes of the DNN and LR EFP classifiers are equal to within error bars, though the overall loss of the DNN is much smaller. A possible interpretation, motivated by Ref. [4], is as follows: since nonlinear combinations of EFPs can be expressed as linear combinations of EFPs of higher degree, the DNN achieves a better loss by learning the optimal nonlinear combinations to effectively incorporate these additional features. However, the marginal returns on data from the set of EFPs are the same, resulting in an identical loss slope. On the other hand, the kNN EMD slope is largest among all classifiers, and distinct from the kNN LOT slope, confirming the fact that these two classifiers are in fact accessing different distance metrics on the space of events. The similarity of the slopes of kNN EMD and the DNN on LOT is remarkable, though it defies a simple geometric explanation and merits further consideration in future work. Finally, it is worth noting that the DNN classifiers have consistently steeper slopes than the kNN classifiers using the same inputs. This may follow from the fact that a marginal datapoint can update the weights of a DNN in way that benefits all test datapoints (or any sharing the same "learned features"), while in a kNN classifier, a new datapoint only benefits test datapoints in a small neighborhood around it.

While we do not optimize over model size for all of our classifiers, in Fig. 3 we consider the effect of varying the number of EFP features used in the logistic regression classifier. In the left panel, we see similar broken power laws for every such choice, with similar slopes

---

[4]The broken power law in the kNN EFP classifier around $T = 50$ is not a result of noise; similar broken power laws can be seen in the LR classifier using various numbers of EFP features in Fig. 3. We do not have a convincing explanation for this behavior and intend to return to it in future work.

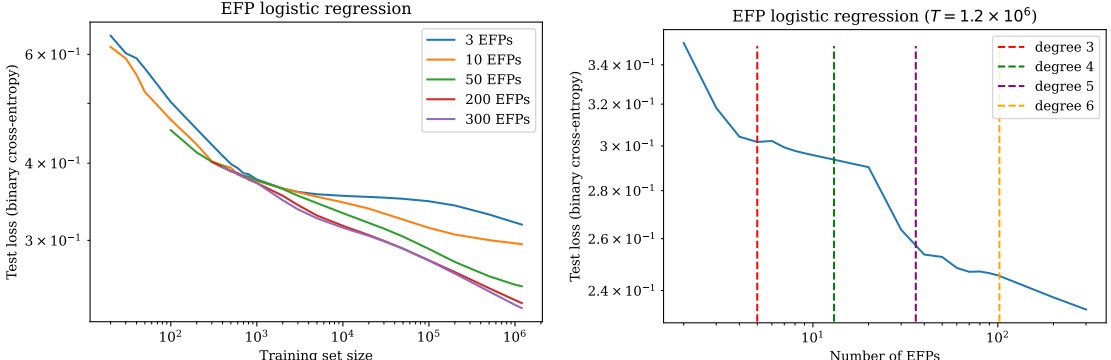

Figure 3: Effect of varying the number of EFP features. As a function of dataset size $T$ (left), different number of EFPs exhibit broken power laws of different slopes. For fixed dataset size (right), breaks in the power law roughly correspond to the inclusion of EFPs of higher degree, which probe smaller angular scales.

before the break but with smaller numbers of features exhibiting shallower power laws after the break. In the right panel, we fix $T$ by using the full training set of $1.2 \times 10^6$ events, and plot the test loss as a function of the number of EFPs. We again see broken power laws, with breaks roughly corresponding to adding EFPs of higher degree. This behavior is reminiscent of the data-feature duality noted in Ref. [4]. Because EFPs of different degrees measure structure in the jet at different physical scales, EFPs are interchangeable only within degree; thus while standard scaling laws predict a single power law as model size or feature number is increased, here power laws only seem to hold within each degree.

However, other aspects of our analysis seem to confound simple interpretations of the scaling laws we observe. Ref. [4] proposed that the scaling exponent of the loss observed in natural data such as large language models or image classification is directly related to a power-law scaling of the empirical data-data covariance matrix,

$$\sigma_{IJ} \equiv \frac{1}{T} \sum_{A=1}^{T} x_{I,A} x_{J,A}, \tag{8}$$

where $A$ indexes the training set and $I, J$ index the features. Since random Gaussian data has a covariance matrix described by the Marchenko-Pastur distribution, which does not have a power-law spectrum, the authors of Ref. [4] hypothesize that neural networks are optimal for processing the power laws of "structured" data (which reflect the correlations among input data at all resolution scales) into power laws in the loss. In particular, they predict that for eigenvalues of $\sigma_{IJ}$ distributed as $\lambda_k \sim \lambda_0 k^{-(1+\alpha)}$, the power-law index for the loss should be $L(T) \propto T^{-\alpha}$.

To test this hypothesis in the context of physics data, Fig. 4 (left) shows the spectrum of the data-data covariance matrix, namely the eigenvalues of $\sigma_{IJ}$ in Eq. (8), for $T = 100$. As anticipated in Ref. [4], the eigenvalues $\lambda_i$ follow a power law spectrum until the small-$\lambda$ tail. However, it is notable that the EFP eigenvalue spectrum for both classes is essentially identical, while the LOT spectrum has the same slope for both classes but differs by a constant. This suggests that power laws in the classifier loss cannot be directly related to power laws in the data spectrum, at least for EFPs. Furthermore, the model of Ref. [4] predicts that steeper slopes for the eigenvalue spectrum correspond to steeper slopes for the loss, while we observe the opposite behavior: LOT has a much shallower eigenvalue spectrum but a steeper classifier slope than all EFP classifiers.

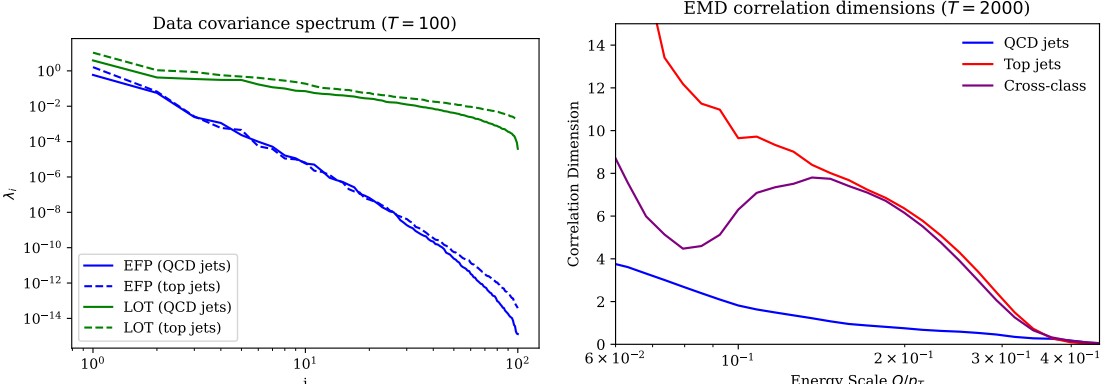

Figure 4: (Left) Spectrum of the data-data covariance matrix for the Euclidean embeddings given by LOT and EFPs, for $T = 100$, where the label $i$ on the $x$-axis indexes the $T$ numerically-ordered eigenvalues from largest to smallest. Power laws are evident in both cases, but the two classes only differ in the tails of the very small eigenvalues for the EFPs, while the power law slopes for LOT are similar but the overall magnitude differs consistently across classes. (Right) correlation dimension as measured by the EMD. Of particular note is the non-monotonic behavior of the cross-class correlation dimension.

Ref. [2] postulates a relation between the loss scaling law and the dimension of the data manifold. However, in our classification problem, the two classes have different intrinsic dimension, which can be traced back the fact that top jets have at least three subjets while QCD jets typically have none. Since the manifold of Lorentz-invariant phase space on which relativistic particles live depends on the number of particles, and the number of observed particles depends in part on the angular resolution scale, the dimension of a jet is not constant but in fact depends on scale. A useful way to measure this is by using the EMD to construct a correlation dimension [25],

$$\dim(Q) = Q \frac{\partial}{\partial Q} \ln \left[ \sum_{i<j} \Theta(Q - \mathrm{EMD}(\mathcal{E}_i, \mathcal{E}_j)) \right], \tag{9}$$

where $\mathcal{E}_i$ and $\mathcal{E}_j$ are jets. This definition computes the logarithmic derivative of the number of events which are within an EMD of $Q$ from each other, and is motivated by the fact that the number of points in a uniformly-sampled ball of radius $Q$ in $\mathbb{R}^d$ grows as $Q^d$. Fig. 4 (right) shows the correlation dimension for QCD jets and top jets from a sample of 2000 events of each class from the test set. The $x$-axis is normalized to the transverse momentum $p_T$ of the jet such that the no events are more than a distance $Q = p_T/2$ away as measured by EMD, so the correlation dimension vanishes at $Q/p_T = 0.5$. As expected, the QCD jet has strictly lower dimension than the top jet, and the dimensions of both classes increase at smaller EMD (corresponding to smaller energy and angular scales, where more jet constituents may be resolved). However, Eq. (9) also permits us to compute a cross-class correlation dimension (as was suggested in Ref. [33] for quark-gluon classification), where we take $\mathcal{E}_i$ from the sample of QCD jets and $\mathcal{E}_j$ from the sample of top jets. As shown in Fig. 4 (right), this dimension is non-monotonic, tracking the top jet dimension at large $Q$ but achieving a local minimum at smaller $Q$ before diverging. It is tempting to interpret this behavior as giving rise to some of the broken power laws we observe, but there appears to be no such break in the kNN EMD classifier, which is the only one that directly uses the EMD. We note that the cross-class correlation dimension measures the extrinsic geometry which is essential for classification problems; the

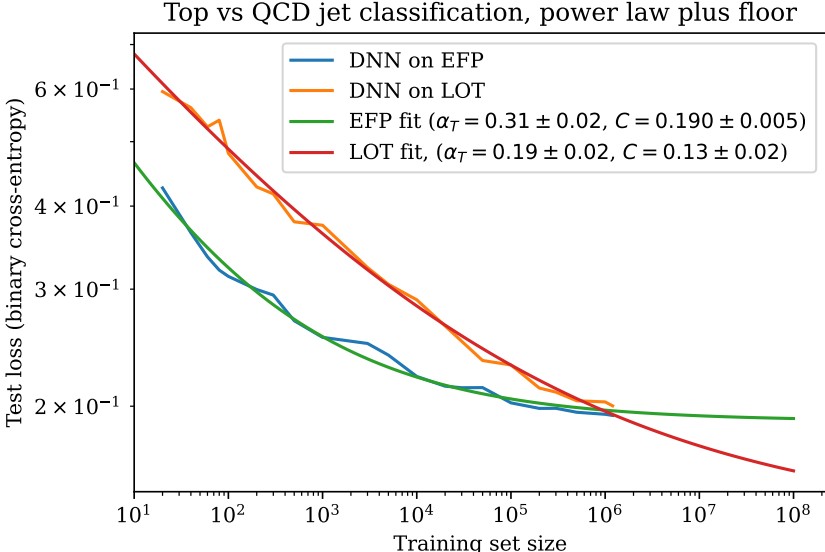

Figure 5: Fitting the DNN classifiers to a power law with floor ($C \neq 0$) yields different values of the irreducible loss $C$. Extrapolating out to $T = 10^8$, this suggests that the LOT classifier may outperform the EFP classifier for sufficiently large $T$, as it has a lower loss floor.

way that the manifolds corresponding to each class are linked or twisted around each other in the embedding space is more important than the dimension of each separately.

Finally, motivated by the softening of the power law slope in the DNN classifiers, we consider a power-law fit which includes a nonzero irreducible loss $C$, using a nonlinear least-squares fit from `optimize.curve_fit` in the `scipy` package. The results of the fit (including the full range of $T$, down to $T = 20$) are shown in Fig. 5, with the fitting functions extrapolated out to $T = 10^8$. Both classifiers appear to be well-fit by such a function, though with much larger slopes $\alpha_T$. Curiously, the ordering of the power law slopes is inverted, with the EFP classifier having a steeper slope, which is obscured somewhat by the EFP classifier's larger irreducible loss. We note that including an irreducible loss $C \neq 0$ for the other four classifiers results in a much poorer fit compared to the case with $C = 0$, with order-1 errors on $C$ for all cases except kNN LOT. We speculate that the poorer fit may be due to the larger overall loss for the non-DNN classifiers, which makes it difficult to identify the approach to the loss floor without a much larger training set to see the continuation of the power law. For the kNN LOT classifier with $T \geq 500$, we obtain $\alpha_T = 0.133 \pm 0.007$ and $C = 0.12 \pm 0.01$. The fact that the irreducible loss is the same to within errors as the DNN classifier trained on LOT suggests that $C$ may be an intrinsic property of the LOT embedding, independent of the classifier.

Taking the scaling laws at face value (both with and without the irreducible loss floor), our results suggest that the DNN LOT classifier may outperform the DNN EFP classifier for datasets larger than the one available from Ref. [11]. In practice, due to the additional uncertainty in these fits, if one wanted to select a classifier for use in online jet tagging, it would certainly be worth extending the training set size an additional decade or more for the two DNN models. In spite of having worse loss at all $T$ measured, the LOT DNN classifier might pull ahead as extrapolated, and it would be interesting and worthwhile to check this prediction on larger datasets.

# 6 Discussion and outlook

In this paper, we have demonstrated the appearance of scaling laws in simulated collider physics data. Across several physically-motivated classifiers, the loss decays as a power law over 3–4 decades of training set size. This behavior is qualitatively similar to scaling laws observed in natural language and image data, but physics data has the advantage of being generated from distributions which are in principle calculable using QFT, and thus one might hope to be able to predict the power law slopes, or at least relationships among them. In this sense, physics could provide a "natural" dataset which could serve as a testbed for theories of manifold learning. The framework of Refs. [20,34–36] for determining a theoretically-optimal classifier in terms of ratios of QFT matrix elements seems suitable for this task, though the challenge will be relating training set size (which is not an observable in QFT) to density of events on the data manifold. Furthermore, such a framework could provide a theoretical prediction for the irreducible loss (see also Ref. [37]), which can be compared with the power law fits including the loss floor. We leave these very interesting aspects of the problem to future work.

For models trained on natural data, neural scaling laws are also observed as a function of compute, which depends both on training data size and model size. The situation we consider here is slightly different, because the compute costs of all of our classifiers (even the neural networks) are dominated by the costs for the EFP, LOT, and EMD preprocessing, which are $\mathcal{O}(T)$, $\mathcal{O}(T)$, and $\mathcal{O}(T^2)$, respectively. We did find that our DNN classifiers were in the data-limited, not parameter-limited, regime, as tripling the model width and depth did not yield any performance gains. Given that collider experiments are effectively in the infinite-data limit, the decision of which classifier to use will be determined in part by compute budget, which suggests a diminishing benefit for using the EMD classifier despite the steeper loss slope. It is important to point out that the *inference* speed of the classifier is actually the limiting factor for online event selection (see e.g. [38]), in which case DNN classifiers (once trained) have an obvious advantage over kNN classifiers since they require only a single forward pass rather than the computation of a $T \times T_{\text{test}}$ distance matrix. From that perspective, even though the kNN classifiers are more directly related to the data geometry, a further systematic analysis of the relationship between model size, training set size, and loss in the DNNs may yield steeper slopes and improved classifier behavior. We leave such a study to future work as well.

Finally, we emphasize that in light of the ubiquity of scaling laws, we would also expect to see them emerge in other jet classification problems such as quark-gluon discrimination, as well as with other classifier architectures. In particular, it would be extremely interesting to see Lorentz-equivariant DNN jet classifier performance, as in Ref. [39], presented as a log-log plot as a function of training set size.[5] While the receiver operating characteristic (ROC) curve is a traditional way of comparing classifier performance, restricting to fixed $T$ does not accurately reflect the abundance of high-energy physics data, and we hope that our work motivates the continued inclusion of scaling law plots in ML applications for physics analyses. Given that collider physics is a quintessential "big data" problem, scaling laws provide an organizing principle to structure investigations into model performance and guide investments in the models which are deployed on real experimental data.

## Acknowledgments

We thank Nathaniel Craig, Katy Craig, Boris Hanin, Andrew Larkoski, David Miller, Ian Moult, Dan Roberts, Dan Carney, Jamie Sully, and Jesse Thaler for stimulating conversations, and Aarav Mande for collaboration in the early stages of this work.

---

[5]Some rough power law behavior appears to be present in the various tasks studied in Ref. [12].

**Funding information** YK thanks the Aspen Center for Physics, which is supported by National Science Foundation grant PHY-2210452, for hospitality during the completion of this work. The work of YK was supported in part by DOE grant DE-SC0015655. This work utilizes the computing resources of the HAL cluster [40] supported by the National Science Foundation's Major Research Instrumentation program, grant #1725729, as well as the University of Illinois Urbana-Champaign. This work also made use of the Illinois Campus Cluster, a computing resource that is operated by the Illinois Campus Cluster Program (ICCP) in conjunction with the National Center for Supercomputing Applications (NCSA) and which is supported by funds from the University of Illinois Urbana-Champaign.

## A   Hyperparameter checks

Here we very briefly illustrate the robustness of our scaling law results to changes in hyperparameters. Fig. 6 (left) shows the result of varying the angular hyperparameter $R$ of the EMD. While $R = 0.8$ matches the optimal transport problem to the jet clustering algorithm, this choice has a larger loss coefficient $A$ than a smaller value of $R = 0.4$, which emphasizes subjet structure more useful for the classification problem at hand. Reducing $R$ too much past this value results in a mismatched angular scale and increases $A$ again.[6] Regardless, the scaling law is identical in all cases, with power law slopes equal to within error bars. Fig. 6 (right) shows the result of changing the size $M_R$ of the reference jet on the kNN LOT classifier. The two loss curves are effectively identical over the entire range of $T$, strongly suggesting that the distinct slope compared to EMD is not an artifact of discretization, but an intrinsic property of the classifier and its distance metric in the continuum limit. Fig. 7 shows the results of cross-validation for selecting the best number of nearest neighbors for the three kNN classifiers, for two different values of $T$. Both EMD and LOT prefer similar numbers of nearest neighbors, as expected from their similar distance metrics, while EFP prefers a larger value.

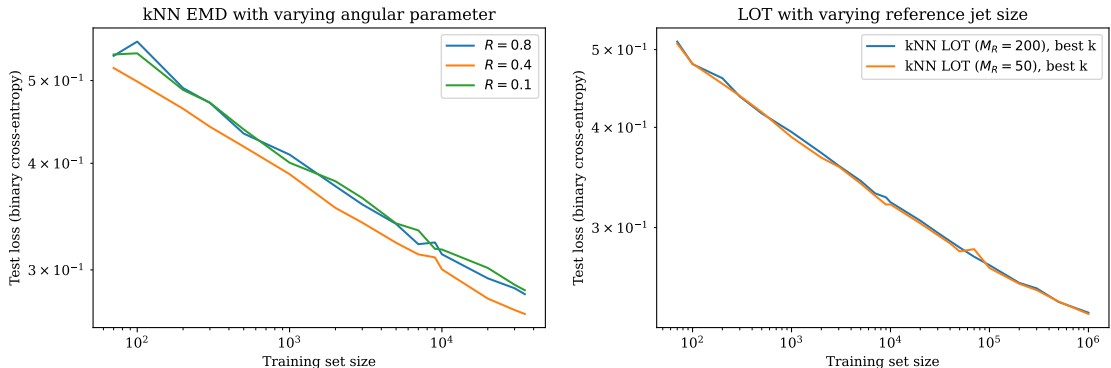

Figure 6: (Left) Varying the angular parameter in the EMD only affects the loss coefficient $A$ but has no effect on the power law slope $\alpha_T$. (Right) Varying the number of particles in the LOT reference jet has no effect on either $A$ or $\alpha_T$.

---

[6]We thank Jesse Thaler for suggesting this interpretation to us.

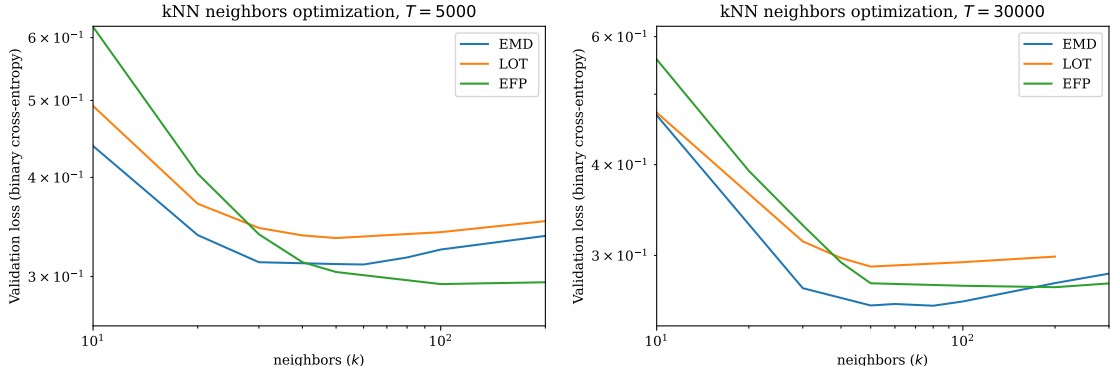

Figure 7: Performance as a function of number of nearest neighbors for the three kNN classifiers we consider, for $T = 5000$ (left) and $T = 30000$ (right).

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
