# Peer review of "Scaling Laws in Jet Classification"

_SciPost Physics Core, doi:SciPost Phys. Core 8, 034 (2025)_

## Round 1 · Referee Report · Anonymous (Referee 1) · 2025-1-21

Report

The article explores scaling laws in jet classification, focusing on the discrimination between top and QCD jets. The authors present a series of classifiers utilizing various architectures and input features, highlighting that all exhibit power-law scaling — a phenomenon widely observed in numerous machine learning applications.

The discussion on power scaling behaviour is compelling. The authors’ conclusion that meaningful comparisons between different classifiers require testing across various training set sizes is particularly relevant for collider physics. However, the methods for data preprocessing and classification draw heavily on established literature, which somewhat limits the article’s originality. Additionally, the interpretation of certain results, especially in Fig. 4, lacks clarity — an issue the authors themselves acknowledge — raising doubts about the generality of the findings.

In summary, I recommend the article for publication in SciPost Physics Core, provided the minor issues listed below are addressed.

Requested changes

  • The significance of the spectrum of the data-data covariance matrix needs to be explained better. How is the spectrum related to the performance of the classifiers? In this context, also the meaning of i as x-label in the left panel of Fig. 4 should be clarified.

  • on page 12: the statement "We note that including C!=0 ... in a much poorer fit." needs to be better explained. Is the "much poorer" referring to the fit for C=0 or in comparison to the fits in Fig. 5. Is there any explanation for the observed worse fits?

  • Regarding Fig. 5: Would it be possible to run one point with an even larger training set size (e.g. 10^7) to test the prediction of the fit curves?

Recommendation

Ask for minor revision

---

## Round 1 · Referee Report · Anonymous (Referee 2) · 2025-2-21

Strengths

The article discusses the occurrence of scaling laws in classification problems in particle physics. The numerical experiments show that different classifiers exhibit an approximate power-law scaling of test loss as a function of training set size with different power-law indices. This result emphasises the importance of comparing classifiers as a function of the size of the data set, rather than for a fixed training set, since the optimal classifier may change significantly as the data set increases.

The work is well presented and comprehensible also for non-particle physicists.

Weaknesses

1) The work is largely exploratory and is based on established methods and data sets. An explanation of the numerical results is missing in many cases - this is of course also due to the complexity of the research question, but reduces the relevance of the results.

2) The discussion of the data covariance matrix and the relevance of the corresponding eigenvalues (see Eq. (8) and Fig. 4) is difficult for non-experts to understand. It is based on A. Maloney, D. A. Roberts and J. Sully, A Solvable Model of Neural Scaling Laws (2022) (Ref [4]); I would appreciate it if the authors could expand this discussion a bit to make the paper self-contained.

Report

Although the paper is mostly exploratory and lacks an explanation of the numerical results, I find the results interesting and the main message, i.e. that classifiers should be compared as a function of the size of the data set, relevant. The paper does not meet the acceptance criteria for SciPost Physics, but I recommend the paper for publication in SciPost Physics Core.

Requested changes

1) Include a brief discussion of the relevance of the data covariance and associated eigenvalues.

Recommendation

Accept in alternative Journal (see Report)

---

## Round 2 · Author Response

List of changes
1. Following comments by Referees 1 and 2, we have added a paragraph on p. 10 (lines 313-323) summarizing the results of Ref. [4] and giving context for the data-data covariance matrix.
2. Following comments by Referee 1, we have added a sentence on p. 12 (lines 368-372) speculating on the poorer fit for including a nonzero loss floor in the non-DNN classifiers.

---

## Round 2 · List of Changes

1. Following comments by Referees 1 and 2, we have added a paragraph on p. 10 (lines 313-323) summarizing the results of Ref. [4] and giving context for the data-data covariance matrix.
2. Following comments by Referee 1, we have added a sentence on p. 12 (lines 368-372) speculating on the poorer fit for including a nonzero loss floor in the non-DNN classifiers.

---

## Editorial Decision

published